# Assessment of knowledge and practice of breast self-examination among reproductive age women in Akatsi South district of Volta region of Ghana

Rita Dadzi[1], Awolu Adam[2,3]*

**1** Department of Epidemiology and Biostatistics, School of Public Health, University of Health and Allied Sciences, Ho, Ghana, **2** Department of Family and Community Health, School of Public Health, University of Health and Allied Sciences, Ho, Ghana, **3** Center for Health Literacy and Rural Health Promotion, Accra, Ghana

* aawolu@uhas.edu.gh

## Abstract

### Background

Breast cancer is the primary cause of cancer death among women globally, responsible for about 425,000 deaths in 2010. This study assessed the awareness, knowledge and practices of breast self-examination as a method of prevention and early diagnosis of breast cancer among reproductive aged women in Akatsi South district in Volta region of Ghana.

### Methods

This study was a cross-sectional study involving 385 women between the ages of 15–49 years. Data were collected with a structured questionnaire and variables included socio-demographic characteristics, breast cancer knowledge, breast self-examination knowledge and practice. Descriptive statistics were used to analyze and present the data and chi square test of significance was used to determine association between socio-demographic variable and practice of breast self-examination.

### Results

The mean age of the women was 24.54±7.19. Only 3.1% of women had no formal education and 58.9% were single. Although 88.3% of the respondents were aware of breast cancer, 64.9% of the respondents had good or sufficient knowledge of breast cancer and only 94 (37.6%) practice BSE. Over 50% of the respondents did not know how to perform BSE. There was a significant association between knowledge on breast cancer and practice of BSE ($\chi2$ = 36.218 p = 0.000). The higher the age of a participant, the lower practice of breast self-examination and this was significant ($\chi2$ = 11.324, p = 0.003).

**Data Availability Statement:** Limited dataset has been deposited to Zenodo and the DOI assigned is (10.5281/zenodo.3526686). The URL is (https://zenodo.org/record/3526686#.XfL3zht7nlV).

**Funding:** The authors received no specific funding for this work.

**Competing interests:** The authors have declared that no competing interests exist.

## Conclusion

Breast self-examination is a key strategy to early detection of breast cancer and subsequently critical for effective treatment and cure of the disease. The findings in this study have shown significant low levels of awareness and practice of breast self-examination among women in Akatsi South district of the Volta region. This pattern may be similar to other rural communities across the region. The need to create awareness and to educate women, especially rural women, on importance of breast self-examination as preventive measure for breast cancer is paramount.

## Introduction

Over the past decade, several research findings and data sources have indicated an increasing burden of breast cancer in terms of incidence, morbidity, and mortality related to breast cancer. Breast cancer is also the primary cause of cancer death among women globally, responsible for about 425,000 deaths in 2010 [1, 2]. It was estimated that in 2012 alone, nearly 1.7 million new cases of breast cancer were diagnosed worldwide (second most common cancer overall after cervical cancer), representing about 12% of all new cancer cases and 25% of all cancers in women [2]. The increasing trend continued in 2018 as the International Agency for Research on Cancer (IARC) estimated that in 2018 alone, there were 2,088,849 new breast cancer diagnosis globally according to data consolidated which constituted 11.6% of all cancers [3]. Again, IARC (2018) reported that there were estimated 626,679 breast cancer related deaths in 2018 alone (ibid). Female breast cancer was the second largest diagnosis of all reported cancer diagnosis in 2018 constituting 11.6% [4].

This trend is alarming and indicative of the global public health threat posed by breast cancer. Of the above diagnosis, Asia and Africa had incidence of 4.8% and 5.8% respectively but mortality estimates were 57.4% and 7.2% for Asia and Africa respectively [2].

As indicated earlier, this hike in breast cancer diagnosis and resultant mortality and morbidity is not limited to particular settings although variations exist in the rates of incidence in different parts of the world. This is because the hikes in breast cancer incidence and prevalence are observed in both developed and developing countries [5]. The estimated mortality for breast cancer rates in the world ranged from 6 to 29 per 100 000, resulting that breast cancer ranks as the fifth cause of death from cancer overall and it is the most frequent cause of cancer death in women in both developing and developed regions [6].

Because of poor access to diagnosis and treatment, women in low- and middle-income countries generally have much poorer outcomes as well. In sub-Saharan Africa only 32% of women are still alive five years after a breast cancer diagnosis, compared with 81% in the USA [7]. In Ghana, breast cancer is a major public health problem and the most common type of cancer among women in terms of mortality and incidence [2]. As far back as 2009, breast cancer was reported as accounting for 16% of all cancers in Ghana and has been a major public health concern [8]. Korle Bu Teaching hospital in Accra and Komfo Anokye Teaching hospital in Kumasi are the two major hospitals in Ghana equipped to treat cancer and were reported to be seeing between 5,000 and 7,000 new cancer patients a year, majority of who were breast cancer patients [9]. This study made sense in the Volta region of Ghana in particular which was reported to have the high breast cancer incidence in a 2016 study of distribution of breast cancer in Ghana [2]. Researchers randomly selected 3,000 women from five regions clinical

breast examination and reported that breast cancer distribution was highest (0.43%) in the Volta region of Ghana[2].

Appropriate preventive strategies focusing on both primary and secondary preventive mechanisms are needed to reduce the incidence of breast cancer. One key strategy is to assess the awareness and knowledge of breast cancer and breast self-examination; and the second major strategy is to help raise breast self-examination among vulnerable women especially those in poor and resource constrained settings. Early identification of breast abnormality is an important step in treating breast cancer and limiting morbidity and mortality caused by breast cancer. Clinical Breast Examination (CBE) and Mammography are now the mainstays for early detection of breast cancer [10]. However, these technologies such as mammography screening are too expensive and out of reach of millions of women around the world and in developing countries in particular [11]. As such for many women in poor resourced settings, regular BSE is a critical strategy in early detection of breast cancer and cure, especially in resource limited settings [12, 13] like Ghana. BSE has a positive effect on the prognosis as well as limits the development of complications and disability, and improves the quality of life and survival. It makes public health sense to continue to recommend BSE because it is free, simple, need low technology and teaching is possible. Monthly BSE is a common method of identifying lumps and other abnormalities in the breast for any signs of abnormality [14]. BSE in conjunction with screening mammography is currently advocated by many organizations, and it is also recommended for younger women starting in their 20's who are not yet being screened by mammography [15]. Breast self-examination as a screening tool for breast cancer in developing countries is advocated in view of its cost-effectiveness [13].

BSE each month between the 7th and 10th day of the menstrual cycle is the simplest yet extremely important way to detect breast cancer at the early stage of growth [ibid]. Doing BSE is one way for a woman to know how her breasts normally feel so that she can notice any changes that do occur. This examination is critical for Ghanaian women because black women have been found to bear the greater burden of breast cancer mortality compared to other races. In order to perform BSE, the individual needs to possess the knowledge of and have the skill of doing so. BSE is important for enabling women become familiar with the feel and appearance of their breast; and help them easily and quickly detect any changes that occur [16]. Women who perform BSE correctly monthly are more likely to detect a lump in the early stage of its tumor development which is critical for successful treatment and survival [ibid]. However, there is evidence that comprehensive knowledge of BSE is still low in many developing countries. For example, in a survey of 790 female household representatives in Southwest Cameroon, only 25% demonstrated adequate knowledge of BSE and only 15% of those with knowledge reported practicing BSE [17]. Lack of awareness of and lack of skill were among reasons BSE were not practiced among Iranian women [18]. In a similar study in Nigeria, researchers reported that while 97.3% of the participants had heard of BSE, only about 50% reported some knowledge of BSE and only 14.5% practiced BSE regularly [19]. Again in Southwest Nigeria, researchers reported that 70% of 425 women surveyed reported not having knowledge and skill to perform BSE [20].

The objectives of the study were to examine awareness and knowledge of breast self-examination as well as the practice of breast self-examination among rural women in Akatsi South district of Volta region of Ghana. In addition, we wanted to explore characteristics or variables that are associated with knowledge and practice of BSE.

## Methods

A descriptive cross-sectional study was conducted whereby 385 reproductive aged women ages 15–49 were conveniently recruited and quantitative data were collected through a

structured questionnaire in Akatsi South district of the Volta region of Ghana. Akatsi South district is predominantly rural with peasant farming and fishing being the main economic activities and an estimated population of 95,426 according the Ghana 2010 Population and Housing census. Demographically, 44,039 were males and 51,387 females constituting 46% and 54% respectively. In order to participate in the study the female had to be between the ages of 15 and 49, and living in Akatsi South district for at least five years at the time of the study. The age range of 15–49 chosen was to ensure that we covered women in reproductive ages. For the residency requirements, we wanted to adhere to the delimitation approved by GHS Ethics Review Committee. We also wanted to make sure that if any issues arose or further research was needed we could trace the participants.

Demographic variables measured in this study included marital status, age, educational attainment, and employment status. Awareness and knowledge of breast cancer were also measured besides the two main variables of knowledge and practice of BSE. To measure awareness of breast cancer among the participants, two questions were included in the questionnaire asking whether or not they had heard of breast cancer as a disease and whether or not they considered breast cancer as a common disease among women in Ghana. The status of socio-demographic variables was important considerations because they determine how independent a woman is in a predominantly male-dominated society. Besides, age and marital status for example have been found to be associated BSE [21]. Association between the demographic variables and knowledge of breast cancer was also measured using $X^2$. Besides reasons provided by those who did not practice BSE, we performed $X^2$ test of significance to determine significant predictors of BSE practice. There were both open-ended and close-ended questions on the questionnaires depending on the variable. For open-ended questions, those who were literate wrote the answers and verbal responses were made by those who could neither read nor write. The questionnaire was written in English and translated in Ewe, the predominant local language in the district, for those who were illiterate. A summary of the questions include breast cancer risk factors, symptoms, and treatment. Also awareness of BSE, skills and when to practice BSE, and whether the participants practiced BSE and when were asked.

Descriptive statistics were used to analyze and present data in percentages and frequency distribution tables and $X^2$ test of significance were performed to determine variables that significantly predicted BSE. For the sake of this study, good knowledge was defined as being to list at least four each of risk factors and symptoms of breast while poor knowledge was defined as not being able to describe at least four each of risk factors and symptoms breast cancer. Informed consent forms had to be signed by adult participants ages 18 and above whilst those between ages 15 and 17 had to sign assent forms after their parents/guardians approved their participation. All those who participated in the study agreed voluntarily to do so prior to recruitment into the study. The study protocol was reviewed and approved by Ghana Health Service Ethics Review Committee (GHS-ERC) with approval number GHS-ERC: 29/05/17. Descriptive statistics were used to analyze and present the data using SPSS version 20.

## Results

In line with the objectives of this study, a number of findings were made. These objectives included assessing knowledge of the rural women about the key symptoms and preventive measures of breast cancer, awareness and knowledge of BSE, and practice of BSE. Analyses of the socio-demographic variables included in the study are first presented. These variables included age, marital status, religious affiliation, occupation, and educational level attained by the participants. Of the 385 respondents, 296(76.9%) were aged less than 30 years, and 17 (4.4%) were 40 years and above. The respondents were between the ages of 15 and 49 years,

with a mean age of 24.54 (±7.19) years. Majority of the respondents (58.9%) were single and never married, 152(39.5%) were married, whilst 1.6% divorced. Majority of the respondents (56.6%) were unemployed, 36.6% reported being self-employed, and 26 (6.8%) were government employees. In terms of educational attainment, majority of the respondents (48.8%) attained high school education whilst 70 (18.2%) attained tertiary education. Only 12 (3.1%) had no formal education. Therefore, the study participants were largely a literate group.

## Awareness and knowledge of breast cancer

The results showed that there was high awareness of breast cancer among women at least in the Akatsi South district. Out of the 385 respondents who participated in the study, majority 340(88.3%) had heard of and were aware of breast cancer as a disease predominantly among women. In terms of prevalence of breast cancer in Ghana, 177(46.1%) of the participants believed breast cancer is most common cancer in women, 106(27.6%) believed it is not common, whilst 101(26.3%) of the respondents did not know how common breast cancer was. As to how they became aware of breast cancer, 154(40%), 115(30%), and 41(10%) of participants mentioned radio, health workers, and school respectively. Other sources of breast cancer information mentioned included friends (8%), neighbors (7%), and books/internet (5%).

To measure knowledge of breast cancer, two sets of questions were presented covering risk factors for breast cancer and symptoms of breast cancer. For risk factors for breast cancer, participants were asked to list at least one risk factor that was likely to expose a woman to breast cancer. The results show that 217 out of the 385 women in this study could not list one risk factor for breast cancer and that constituted 56.3% well over of half of the study sample. Risk factors listed included family history, contraceptive use, sex, not breastfeeding, menopause, obesity or overweight and older age. Fifty-two people (12%) each mentioned family history and women who did not breastfeed as risk factors for breast cancer. Contraceptive use, being a woman, and obesity or overweight were listed by 37 (8.6%), 31 (7.2%), 18 (4.2%) were mentioned respectively. Other risk factors listed were effects of menopause and increasing age which constituted 3.2% and 2.8% respectively. The women were then asked whether breast cancer is curable. Most of them 244(63.6%) reported that they believed breast cancer is curable if detected early. Only 42(10.9%) of them believed that breast cancer is not curable but can be controlled. Like the risk factors, participants were asked to list at least one symptom of breast cancer known of or they had heard of. Pain in the breast was the most listed whereby 133 women constituting 26.8% mentioning it as a major symptom of breast cancer. This was followed by lump in the breast which was listed by 110 (22%) women. Sudden swollen or thickness of the breast, change in size or shape of breast, redness of nipple, and nipple discharge other than breast milk are correct answers and were mentioned by 87 (17.5%), 82 (16.5%), 44 (8.9%), and 41 (8.3%), respectively.

Therefore, while awareness about breast cancer was relatively high in this study population (88%), knowledge was basic and not adequate with 56% not knowing any risk factor for breast cancer, knowledge of symptoms was moderate.

Respondents were assessed on knowledge of breast cancer by computing knowledge variables to determine average knowledge score. Series of questions were asked with different answer choices and the number of correct answers or scores provided by a participant showed his or her knowledge level. The mean score was 0.64 (64%). Respondents who performed below the average score were classified as having poor knowledge and those who had average or above were classified as having good knowledge on breast cancer. Out of the 385 respondents interviewed, majority 65% had good knowledge about breast cancer whilst, 35.1% of them had poor knowledge about breast cancer.

## Association between knowledge of breast cancer and demographic characteristics

Age, marital status, occupation, education level and religion were cross tabulated against knowledge to determine their statistical significance. Out of the 135 respondents that had poor knowledge, majority of them 104(77.0%) were aged less than 30 years and 25(18.8%) were aged between 30 and 39 years. Only 6(4.4%) were aged above 40 years. Among the 250 respondents who had good knowledge, majority were aged less than 30 years. There was no significant relationship between a respondent age and level of knowledge ($\chi2 = 0.004$, p = 0.998).

Majority (57.8%) of the respondents who had poor knowledge were single and 55(40.7%) were married. Only 2(1.5%) were divorced. Also, among those that had good knowledge, majority 149 (59.6%) were single and 97(76.8%) were married and Only 4(1.6%) were divorced. There was no significant relationship between a respondent's knowledge and marital status ($\chi2 = 0.14$, p = 0.932).

Only 4(3.0%) of the respondents who had poor knowledge were employed whilst, majority 67(49.6%) of them were unemployed and 64(47.4%) were self-employed. Among the respondents who had good knowledge, only 22(8.8%) of them were employed whilst, majority 151 (60.4%) of them were unemployed and 77(30.8%) were self-employed. There was significant relationship between a respondents occupation and level of knowledge ($\chi2 = 12.82$, p = 0.002).

On the other hand, majority 61(45.2%) of the respondents who reported poor knowledge attained primary/Junior High School (JHS) and 56(41.5%) attained Senior High School (SHS). Only 12(8.9%) of them attained tertiary and 6(2.4%) had no education. Also, among respondents that had good knowledge, majority 132(52.8%) of them attained SHS/SSS and 58(23.2%) attained tertiary. Only 6(2.4%) had no education and 54(21.6%) attained Primary/JHS. There was a significant association between participant level of education and knowledge ($\chi2 = 29.68$, p = 0.000). Table 1 shows the association between knowledge about breast cancer and demographic characteristics.

## Knowledge and practice of breast self-examination

BSE knowledge and practice assessment were the major objectives of this study. This study aimed to determine the proportion of respondents who were aware of and had comprehensive knowledge of and skill to perform BSE and are practicing BSE. The findings show that knowledge of BSE was generally low among the 385 women who participated in this study. Only 165 (43.3%) of the respondents knew what self-breast examination was whilst majority 217(56.7%) reported not knowing anything about it or having not heard about it. Out of the 385 respondents surveyed, majority 235 (61.0%) reported not knowing anything about BSE and were never taught how to perform it. Among those who reported they learnt about BSE, most 90 (60.1%) of them were taught by a health worker, 17(11.3%) were taught by relative or friends. Fourteen (9.3%) learnt the procedure of BSE through books and 29(20.3%) learnt it through other means. Only 39(36.8%) of the respondents believes the most appropriate time for self-breast examination is between 2 and 3 days after cessation of menstruation. However, majority 49(46.2%) thought BSE can be carried out any time within the month. Seven (6.6%) of them thought the appropriate time for self-breast examination was a few days before menstruation starts.

In terms of BSE practice, only 106 (27.5%) of the respondents in this study reported practicing or ever practiced breast self-examination. Overwhelming majority 279 (72.5%) reported not practicing BSE. Among those who practiced BSE, most 62(58.5%) reported performing BSE every month, 27(25.5%) examined the breast once every month and about 17(16%) examine the breast once every year. These participants provided various reasons and circumstances

**Table 1. Demographic variables and their association to knowledge of breast cancer.**

| Variable | Overall Knowledge | | Chi-square (χ2) | P-value |
|---|---|---|---|---|
| Age Group | Poor knowledge [N = 135] | Good knowledge [N = 250] | | |
| <30 | 104(77.0) | 192(76.8) | | |
| 30–39 | 25(18.5) | 47(18.8) | | |
| 40+ | 6(4.4) | 11(4.4) | 0.004 | 0.998 |
| Marital Status | | | | |
| Single | 78(57.8) | 149(59.6) | | |
| Married | 55(40.7) | 97(38.8) | | |
| Divorced | 2(1.5) | 4(1.6) | 0.14 | 0.932 |
| Religion | | | | |
| Christian | 125(92.6) | 244(97.6) | | |
| Muslim | 6(4.4) | 3(1.2) | | |
| Traditionalist | 3(2.20 | 1(0.4) | | |
| Others | 1(0.7) | 2(0.8) | 6.98 | 0.072 |
| Occupation | | | | |
| Employed | 4(3.0) | 22(8.8) | | |
| Unemployed | 67(49.6) | 151(60.4) | | |
| Self Employed | 64(47.4) | 77(30.8) | 12.82 | 0.002 |
| Education | | | | |
| None | 6(4.4) | 6(2.4) | | |
| Primary/JHS | 61(45.2) | 54(21.6) | | |
| SHS/SSS | 56(41.5) | 132(52.8) | | |
| Tertiary | 12(8.9) | 58(23.2) | 29.68 | 0.000 |

that hindered them from practicing BSE. Out of the 279 (72.5%) who do not perform breast self-examination, more than half 50.1% of them said they don't know the techniques for breast self-examination. Some (17.4%) respondents also said they do not practice BSE because they don't have breast cancer problems. Lack of privacy at home was also mentioned by 7.3% of the respondents as reason for not practicing BSE. About 3.4% of the respondents believed they did not need to self-examine their breast and another 3.4% did not feel comfortable to self-examine their breast. About 5.7% of them also gave other reasons for not practicing BSE. The results are presented in Table 2 below.

## Predictors of breast self-examination

The age of a woman was a strong predictor of BSE practice among this study population. There was a significant association between age group and practice of breast self-examination (χ2 = 11.32, p = 0.003). A higher age of a participant corresponded with a lower practice of BSE. Among respondents who practice BSE, most 70(66.0%) of them were aged less than 30 years, followed by 30–39 27(25.5%). Therefore, younger women below age 30 were more likely to perform BSE compared to women over the age of 40. Another predictor of BSE practice established among the study population was knowledge of breast cancer itself. A series of questions were asked to determine the general awareness and knowledge of breast cancer and 250 (65%) of the respondents reported good knowledge of breast cancer. Of the 250, 94 (37.6%) reported practicing BSE. On the other hand, 135 (35%) reported poor knowledge of breast cancer and of those 135 respondents, only 12 reported practicing BSE. To confirm knowledge

**Table 2. Knowledge and practice of breast self-examination.**

| Variable | Frequency | Relative Percent (%) |
|---|---|---|
| Are you aware of or ever heard of Breast Self-Examination? | | |
| Yes | 167 | 43.3 |
| No | 217 | 56.7 |
| Source of BSE information | | |
| Health worker | 90 | 60.1 |
| Family member | 17 | 11.3 |
| Books/Print material | 14 | 9.3 |
| Others | 29 | 20.3 |
| Frequency for BSE | | |
| Monthly | 60 | 58.5 |
| Every 6 months | 21 | 25.5 |
| Annually | 17 | 16 |
| Skills to Perform BSE | | |
| Yes | 150 | 39 |
| No | 235 | 61 |
| Appropriate Age to Perform BSE | | |
| Before age 20 | 50 | 47.2 |
| 20–25 | 33 | 31 |
| 26–30 | 14 | 13.2 |
| 35+ | 9 | 6.5 |
| Appropriate time to perform BSE | | |
| 2–3 days after cessation of menstruation | 39 | 36.8 |
| Monthly on a fixed day | 11 | 10.4 |
| Few days before menstruation starts | 7 | 6.6 |
| Any time within the month | 49 | 46.2 |
| Practice of BSE | | |
| Yes | 106 | 27.5 |
| No | 279 | 72.5 |

of breast cancer as a significant predictor, a $X^2$ was performed and significant association ($\chi2 = 36.218$ p = 0.000) found.

## Discussion

The discussion highlights the main findings of this study and how those findings compare with findings from similar studies conducted on the subject of breast cancer and breast self-examination. In this current study breast cancer awareness was high for the rural population in that 88.3% of the respondents were aware of breast cancer but only 64.9% of the respondents demonstrated adequate knowledge of symptoms and preventive measures for breast cancer. In terms of awareness of BSE, only 43.3% reported being aware of BSE in the current study. This finding of low awareness and knowledge of BSE is similar but lower compared to the findings in Nigeria [19] in which 97% reported ever hearing of BSE and only about 50% reported adequate knowledge of BSE. It is also similar to a finding of poor knowledge of BSE in Ibadan, Southwest Nigeria where 70% of 603 market women, that is women buying and selling various commodities in the market, reported not knowing about and how to perform BSE. Earlier studies reported similar poor knowledge of BSE [20,22].

Another major objective in this study was to examine the practice of breast self-examination among the women. The findings in this study show that only 106 (27.5%) of the 385 women reported practicing or ever practiced BSE. Two hundred and seventy-nine (279) women reported they never practiced BSE which constituted 72.5% of the sample in this study. This finding is consistent with the finding of similar studies in other parts of the world whereby the practice of BSE is generally low. For example, in a study in Cameroon, only 15% of 200 women who reported basic knowledge of BSE actually practiced BSE [16] and in Oyo State in Nigeria, 14.5% reported engaging in BSE [19]. The finding was lower when compared to findings in relatively more developing countries like in India and Malaysia where BSE practice was measured at 59% [22] and 55% [23], respectively.

Two variables were found to be significantly associated with and predicted practice of BSE including knowledge of breast cancer and age of the woman. This current study reported a significant association between knowledge of breast cancer and practice of BSE ($\chi2$ = 36.218 p = 0.000). In this current study, out of the 250 (65%) of the respondents with good knowledge about breast cancer, 94(37.6%) practiced BSE. This finding was consistent with what was reported by Dundar and colleagues who found level of breast cancer knowledge to be the only variable significantly associated with the BSE and mammography practice (p = 0.011, p = 0.007) [20]. Other researchers also found that performing breast self-examination is significantly related to knowledge of breast cancer and knowledge about breast cancer screening programs (p < 0.05) [24].

This study also showed a significant association between age group and practice of breast self-examination ($\chi2$ = 11.324, p = 0.003). As the age increased the level of participant practice of BSE was lower. Among respondents who reported practicing BSE, most 70(66.0%) were aged less than 30 years, followed by 30–39 years 27(25.5%), and 40 years and above 9(8.5%). This finding is also consistent with other findings [25, 24, 23]. The association between age and BSE may be due to younger women being more involved in social events and media including watching television, Facebook, Whatsup, and YouTube. In addition, younger women are in formal education today than older women. Younger women are therefore, more exposed to information about health issues than older women.

Among those who reported not practicing BSE, many reasons were given to the low practice. For instance, half (50.1%) of the respondents said they did not know the techniques for breast self-examination. Some respondents (17.4%) also said they did not practice BSE because they didn't have breast cancer problems. Lack of privacy at home was also mentioned by 7.3% of the respondents as reason for not practicing BSE. About 3.4% of the respondents did not think they need to self-examine their breasts and another 3.4% said they did not feel comfortable to self-examine their breast. These findings were consistent with other researchers who stated that, regarding the barriers to BSE, the majority who never practiced BSE mentioned that lack of knowledge, not having any symptoms, and being afraid of a breast cancer diagnosis were the main barriers to practicing BSE (20.3%; 14.3%; 4.4% respectively) [24]. It is important to note that the socio-demographic variables measured in this study were crucial in understanding the responses of the participants and the findings in this study. It is important to note that there a number of limitations during the conduct of this study. One limitation was a rush in completing the questionnaire as many of the participants complained of busy schedules and not having much to spend. There was possibility that some participants may not have fully understood the questions prior to answering the questions. The second limitation was translating the questionnaire to Eve, the local language predominantly spoken in the study location. We had to get translators for those who could neither read nor write English. We felt that the translators may not have fully translated the exact questions to the participants who were illiterates. Finally, a convenience sampling technique was employed to recruit participants into

the study which a weaker compared to random sampling. Despite the above limitations, every effort was made to abide by and to ensure that all ethical standards and requirements in conducting human subjects research, especially voluntary participation by participants. Besides, most of the questionnaires were fully completed by the participants and not much data cleaning was needed.

## Conclusion

In conclusion, the findings in this study show an overall lack of comprehensive knowledge of breast cancer and BSE among rural women in the Volta region of Ghana. Also practice of BSE, which is an important method for early diagnosis of breast cancer especially in resource limited settings like the Volta region of Ghana, is very low. The findings did show that knowledge of breast cancer and BSE as well as age of a woman are strong predictors of BSE. The implication for public health practice is that there is the need to develop and/or adopt culturally appropriate and proven educational and skill building interventions to inform and train rural women on BSE targeting all women. There is the need to improve knowledge of breast cancer and BSE and to target all age groups in future breast cancer awareness programs.

## Acknowledgments

We wish to sincerely thank all those who supported us in the preparation and data collection during this study. Particular thanks to Patrick Azooya of the University of Health and Allied Sciences, who helped in questionnaire printing and organization.

## Author Contributions

**Conceptualization:** Rita Dadzi.

**Data curation:** Rita Dadzi.

**Formal analysis:** Rita Dadzi, Awolu Adam.

**Investigation:** Rita Dadzi.

**Methodology:** Awolu Adam.

**Resources:** Rita Dadzi, Awolu Adam.

**Software:** Rita Dadzi.

**Supervision:** Awolu Adam.

**Validation:** Awolu Adam.

**Writing – original draft:** Rita Dadzi.

**Writing – review & editing:** Awolu Adam.

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
