## [Decision Letter · Decision Letter 0]

26 Sep 2019

PONE-D-19-20459

Assessment of Knowledge And Practice Of Breast Self-Examination Among Reproductive Age Women in Akatsi South District of Volta Region of Ghana.

PLOS ONE

Dear Dr. ADAM,

Thank you for submitting your manuscript to PLOS ONE. After careful consideration, we feel that it has merit but does not fully meet PLOS ONE’s publication criteria as it currently stands. Therefore, we invite you to submit a revised version of the manuscript that addresses the points raised during the review process.

**Abstract:**

Background- replace “(district or township name)” with the actual location informationMethods- the software used for data analysis are probably the least important details to include. Instead include information on which variables were measured and the statistical analysis. The abstract says that women aged 15-45 were included but the rest of the paper indicates that women aged 15-49 were included.Conclusion- Your text states that breast self-examination is key to early detection, effective treatment and cure of breast cancer. This wording is too strong and I suggest tempering it to say instead that it is A key strategy.

**Introduction:**

The first two sentences of the introduction are unclear and should be clarified.The introduction is too long and repetitive and should be shortened and streamlined.In the paragraph that starts “Because of poor access…” the sentence “Korle Bu Teaching Hospital…” clarify if these hospitals are in Ghana. In the next sentence, clarify that Volta is in Ghana. In the last sentence clarify whether the five regions you refer to are in Ghana.Remove the heading in the introduction section “Awareness of breast self-examination and breast cancer”Resource is misspelled as resourseStatement “Breast self-examination is critical in resource limited settings as a strategy to reducing mortality due to late reporting and diagnosis” requires a referenceStatement “Breast selfexam (BSE), or regularly examining your breasts on your own, can be an important way to find a breast cancer early, when it's more likely to be treated successfully.” requires a referenceStatement “Breast selfexamination as a screening tool for breast cancer in developing countries is advocated in view of its cost-effectiveness.” requires a referenceThe research objective “The objectives of the study were therefore, to examine awareness and knowledge of breast selfexamination as well as the practice of breast self-examination among rural women in Akatsi South District of Volta Religion of Ghana” does not totally match the results you present. Please rewrite your objective to reflect the fact that you are exploring which characteristics are associated with knowledge and BSE

**Methods:**

Your methods section is missing quite a bit of information.  Please add:Information about the setting. The district will, of course, not be familiar to readers outside Ghana. What type of setting is this? Rural? Urban? How populated?Please justify why you chose to include women ages 15-49 in your studyPlease justify why you included only women who had resided in the study area for 5 years or morePlease explain what variables are measured in this study (marital status, religion, etc) and justify why each is included. For example, is there existing evidence that religious affiliation is associated or might be associated with your two main variables in this setting?Please enumerate whether the questions asked were open ended or multiple choice. For example, was the question about risk factors for breast cancer open ended, where women had to write in answers or verbally respond without any cues? Or did they choose from a list?You refer to “good” knowledge and “poor” knowledge in your results. Please provide a justification for labeling knowledge good or poor.Explain what statistical analysis you performed

**Results:**

The sentence “These socio-demographic variables were crucial in understanding the responses of the participants and the findings in this study.” should be moved to the discussion as it is a commentary on your findings.The sentence “To measure awareness of breast cancer among the participants, two questions were included in the questionnaire asking whether or not they had heard of breast cancer as a disease and whether or not they considered breast cancer as a common disease among women in Ghana.” belongs in the methods.I suggest changing the phrase “refusal to breast feed” to “did not breast feed” in the following sentence since some women might not have given birth and some women may have been unable to physically breast feed. “Fifty-two people each mentioned family history and refusal to breastfeed as risk factors for breast cancer and that constituted 12% each.”Not all of your readers will be experts on breast cancer. You should explicitly state whether the responses given by participants (e.g. list one symptom of breast cancer) were correct.Avoid the word “influence” in your results because this implies a causal association. (“After the knowledge of breast cancer was determined above as to either good or poor knowledge, it was important to find out the demographic characteristics of respondents that may have influenced their level of knowledge.”) Also, this sentence really belongs in the methods.Please revise the title for Table 2 so that it is more informative and clearly and specifically describes the information in the tableThis sentence belongs in the methods: “Even though those who reported not practicing BSE gave various reasons for not practicing, we wanted to statistically find out the predictors of BSE in this sample and so a chi square (X 2 ) test of significance was performed.”

**Discussion:**

Please explain what you mean by “market” womenPlease provide an explanation of why age may relate to BSE. You do provide some references. Has there been any discussion or do you have any ideas about why this relationship exists?Limitations and strengths need to be added to your discussion sectionApplications of your research findings should be suggested

**General comments: **

Data are plural. Please adjust the text throughout.Avoid using the language “increase” and “decrease” throughout and instead say “higher” or “lower.” The words increase and decrease imply a longitudinal association across time. You are describing cross-sectional associations.

We would appreciate receiving your revised manuscript by Nov 10 2019 11:59PM. To enhance the reproducibility of your results, we recommend that if applicable you deposit your laboratory protocols in protocols.io, where a protocol can be assigned its own identifier (DOI) such that it can be cited independently in the future. For instructions see: http://journals.plos.org/plosone/s/submission-guidelines#loc-laboratory-protocols

We look forward to receiving your revised manuscript.

Kind regards,

Carmella August

Academic Editor

PLOS ONE

Journal Requirements:

2. We ask that you please include the reference number for your ethics approval in your ethics statement.

Reviewers' comments:

Reviewer's Responses to Questions

**Comments to the Author**

1. Is the manuscript technically sound, and do the data support the conclusions?

Reviewer #1: Yes

Reviewer #2: Yes

2. Has the statistical analysis been performed appropriately and rigorously? 

Reviewer #1: Yes

Reviewer #2: Yes

3. Have the authors made all data underlying the findings in their manuscript fully available?

Reviewer #1: No

Reviewer #2: Yes

4. Is the manuscript presented in an intelligible fashion and written in standard English?

Reviewer #1: Yes

Reviewer #2: Yes

5. Review Comments to the Author

Reviewer #1: Introduction

Authors should refer to GLOBOCAN 2012 as reference 1. As the manuscript Ghartey et al refers to Ghanaian specific breast cancer distribution.

The section on BSE of the introduction is repetitive and verbose. It would serve the authors to reduce the length and be more concise.

Methods

Need to be listed in methods:

A summary of the questions asked, which language or languages (s) etc. Were the questions on scales, yes/no, categorical. What variables were collected from study participants. Where were the study participants identified?

Results,

Is the entire region rural or are there more urban centers within the region? Can the authors analyze based on this category.

Reviewer #2: This is a timely article. The topic is good and the data connect to the discussion at hand. It is appreciated that the significant and non-significant findings are presented. Edits would include using past tense throughout and clarifying some of the statements. I have uploaded a scanned copy of my comments.

6. PLOS authors have the option to publish the peer review history of their article (what does this mean?). If published, this will include your full peer review and any attached files.

Reviewer #1: No

Reviewer #2: No

---

## [Author Response · Author response to Decision Letter 0]

7 Nov 2019

We would like to sincerely thank the reviewers for the meticulous review they conducted. We have laerned and have grown from the review and have tried to address all the comments to the best of ability. We hope that our revision makes the manuscript even more meritorious for acceptance and publication.

---

## [Editor Report · Decision Letter 1]

14 Nov 2019

PONE-D-19-20459R1

Assessment of Knowledge And Practice Of Breast Self-Examination Among Reproductive Age Women in Akatsi South District of Volta Region of Ghana.

PLOS ONE

Dear Dr. ADAM,

Thank you for submitting your manuscript to PLOS ONE. After careful consideration, we feel that it has merit but does not fully meet PLOS ONE’s publication criteria as it currently stands. Therefore, we invite you to submit a revised version of the manuscript that addresses the points raised during the review process.

We would appreciate receiving your revised manuscript by Dec 29 2019 11:59PM. To enhance the reproducibility of your results, we recommend that if applicable you deposit your laboratory protocols in protocols.io, where a protocol can be assigned its own identifier (DOI) such that it can be cited independently in the future. For instructions see: http://journals.plos.org/plosone/s/submission-guidelines#loc-laboratory-protocols

We look forward to receiving your revised manuscript.

Kind regards,

Carmella August

Academic Editor

PLOS ONE

Additional Editor Comments (if provided):

Thank you for responding to my comments on the first version of your paper. The paper is much improved. I have listed my additional concerns below, please respond and PLEASE INDICATE LINE NUMBERS WHERE YOU MADE CHANGES so that I can easily find your changes. Thank you.

My comments:

1. I asked you to change the conclusion in your abstract to "a key strategy" and you changed it to "key strategy." Please add the "a" so it reads "a key strategy."

2. Your methods is one large paragraph. Please split the text into multiple paragraphs.

3. I cannot find the revised text that you mention in your response #10 in the introduction about BSE, or regularly examining your breasts on your own... Can you please provide line numbers so that I can easily find where you made this change?

4. For your response #18 in the results section of your response to me, I can't find where you made this change. Please respond and include line numbers to make it easy for me to find this change.

Reviewer 1 comments:

1. Please respond to reviewer 1's comments asking about analyzing your data based on geography.

Reviewer 2 comments:

Please review and apply edits made by reviewer 2 in the scanned file provided.

Thank you.

Ella August

---

## [Author Response · Author response to Decision Letter 1]

27 Nov 2019

Reviewer #1: Introduction

1. Authors should refer to GLOBOCAN 2012 as reference 1. As the manuscript Ghartey et al refers to Ghanaian specific breast cancer distribution.

Response: Ferlay J, Soerjomataram I, Dikshit R, Eser S, Mathers C, Rebelo M, Parkin DM, Forman D, Bray F. Cancer incidence and mortality worldwide: sources, methods and major patterns in GLOBOCAN 2012. International journal of cancer. 2015 Mar 1;136(5):E359-86.

The above reference has been provided as reference. Ferlay and colleagues worked on GLOBOCAN 2012 and is an appropriate reference on GLOBOCAN 2012.

2. The section on BSE of the introduction is repetitive and verbose. It would serve the authors to reduce the length and be more concise.

Response: The BSE section has been cut down and well synchronized.

3. Methods: Need to be listed in methods:

A summary of the questions asked, which language or languages (s) etc. Were the questions on scales, yes/no, categorical. What variables were collected from study participants. Where were the study participants identified?

Response: The summary of the questions included breast cancer and breast self-examination knowledge and BSE practice. The variables collected from the participants are clarified in the methods section on pages 7 and 8 and lines 148 to 155.

4. Results: Is the entire region rural or are there more urban centers within the region? Can the authors analyze based on this category.

Response: Except for Ho, the capital town of the Volta Region of Ghana, which is urban, the rest of the region is predominantly rural. Akatsi South district is predominantly rural and only women from the district were recruited into the study. Therefore, the data analysis reflects predominantly rural women and we do not have urban data to analyze in this study. 

Reviewer #2: This is a timely article. The topic is good and the data connect to the discussion at hand. It is appreciated that the significant and non-significant findings are presented. Edits would include using past tense throughout and clarifying some of the statements. I have uploaded a scanned copy of my comments.

Response:

All the edits recommended by the reviewer 2 have been effected throughout the manuscript. We did not feel the need to track all the edits with track changes.

---

## [Editor Report · Decision Letter 2]

10 Dec 2019

Assessment of Knowledge And Practice Of Breast Self-Examination Among Reproductive Age Women in Akatsi South District of Volta Region of Ghana.

PONE-D-19-20459R2

Dear Dr. ADAM,

We are pleased to inform you that your manuscript has been judged scientifically suitable for publication and will be formally accepted for publication once it complies with all outstanding technical requirements.

With kind regards,

Carmella August

Academic Editor

PLOS ONE
---

## [Editor Report · Acceptance letter]

19 Dec 2019

PONE-D-19-20459R2 

Assessment of Knowledge And Practice Of Breast Self-Examination Among Reproductive Age Women in Akatsi South District of Volta Region of Ghana. 

Dear Dr. ADAM:

I am pleased to inform you that your manuscript has been deemed suitable for publication in PLOS ONE. Congratulations! Your manuscript is now with our production department. 

With kind regards,

on behalf of

Dr. Carmella August 

Academic Editor

PLOS ONE